# Validation of the CLIF-C OF Score and CLIF-C ACLF Score to Predict Transplant-Free Survival in Patients with Liver Cirrhosis and Concomitant Need for Intensive Care Unit Treatment

**DOI:** 10.3390/medicina59050866

**Published:** 2023-04-29

**Authors:** Michael Nagel, Ruben Westphal, Max Hilscher, Peter R. Galle, Jörn M. Schattenberg, Oliver Schreiner, Christian Labenz, Marcus Alexander Wörns

**Affiliations:** 1Department of Internal Medicine I, University Medical Center of the Johannes Gutenberg-University, 55131 Mainz, Germany; 2Cirrhosis Center Mainz (CCM), University Medical Center of the Johannes Gutenberg-University, 55131 Mainz, Germany; 3Department of Gastroenterology, Hematology, Oncology and Endocrinology, Klinikum Dortmund, 44137 Dortmund, Germany; 4Institute of Medical Biostatistics, Epidemiology and Informatics, University Medical Center of the Johannes Gutenberg-University, 55131 Mainz, Germany

**Keywords:** liver cirrhosis, endstage liver disease, acute-on-chronic liver failure

## Abstract

Both the Chronic Liver Failure Consortium (CLIF-C) organ failure score (OFs) and the CLIF-C acute-on-chronic-liver failure (ACLF) score (ACLFs) were developed for risk stratification and to predict mortality in patients with liver cirrhosis and ACLF. However, studies validating the predictive ability of both scores in patients with liver cirrhosis and concomitant need for intensive care unit (ICU) treatment are scarce. The aim of the present study is to validate the predictive ability of the CLIF-C OFs and CLIF-C ACLFs regarding the rationale of ongoing ICU treatment and to investigate their predictive ability regarding 28-days (short-), 90-days (medium-), and 365-days (long-term) mortality in patients with liver cirrhosis treated in an ICU. Patients with liver cirrhosis and acute decompensation (AD) or ACLF and concomitant need for ICU treatment were retrospectively analyzed. Predictive factors for mortality, defined as transplant-free survival, were identified using multivariable regression analyses and the predictive ability of CLIF-C OFs, CLIF-C ACLFs, MELD score, and AD score (ADs) was assessed by determining the AUROC. Of 136 included patients, 19 patients presented with AD and 117 patients with ACLF at ICU admission. In multivariable regression analyses, CLIF-C OFs as well as CLIF-C ACLFs were independently associated with higher short-, medium-, and long-term mortality after adjusting for confounding variables. The predictive ability of the CLIF-C OFs in the total cohort in short-term was 0.687 (95% CI 0.599–0.774). In the subgroup of patients with ACLF, the respective AUROCs were 0.652 (95% CI 0.554–0.750) and 0.717 (95% CI 0.626–0.809) for the CLIF-C OFs and for the CLIF-C ACLFs, respectively. ADs performed well in the subgroup of patients without ACLF at ICU admission with an AUROC of 0.792 (95% CI 0.560–1.000). In the long-term, the AUROCs were 0.689 (95% Cl 0.581–0.796) and 0.675 (95% Cl 0.550–0.800) for CLIF-C OFs and CLIF-C ACLFs, respectively. The predictive ability of CLIF-C OFs and CLIF-C ACLFs was relatively low to predict short- and long-term mortality in patients with ACLF with concomitant need for ICU treatment. However, the CLIF-C ACLFs may have special merit in judging futility of further ICU treatment.

## 1. Introduction

Despite advances in the management and treatment of chronic liver diseases, liver cirrhosis remains the fourth leading cause of death in Europe, and the mortality from liver cirrhosis at a global scale is increasing [1]. Especially, decompensated liver cirrhosis requiring hospitalization is associated with a high short-term mortality. Particularly, patients with liver cirrhosis and acute-on-chronic liver failure (ACLF) have the worst short-term prognosis. In recent years, ACLF has been redefined as a distinct syndrome including decompensated liver cirrhosis and the presence of extrahepatic organ failure [2]. The development of ACLF is often triggered by sudden events such as infections or gastrointestinal bleeding and the syndrome is characterized by a severe systemic inflammatory response syndrome (SIRS) with an increased release of cytokines [3]. Prognosis of these patients is mainly determined by the presence and the degree of extrahepatic organ failure. In the CANONIC study of the European Association for the Study of the Liver (EASL), chronic liver failure consortium (CLIF), several prognostic factors were identified in a large cohort of patients with decompensated liver cirrhosis. Consequently, a simplified organ function scoring system (CLIF-organ failure score [CLIF-C OFs]) was developed for the diagnosis and grading of ACLF and subsequently, CLIF-C OFs and two other independent predictors of mortality (age and white blood cell count [WBC]) were combined to develop a specific prognostic score for ACLF (CLIF-C ACLF score [CLIF-C ACLFs]) [4]. In this study, especially patients with high CLIF-C ACLFs (≥65 points) had a very poor prognosis with a 28-days mortality rate > 80% in the validation set. At present, no therapeutic intervention that influences survival outside of supportive care, e.g., treatment on an intensive care unit (ICU), antibiotics, fluid resuscitation, or vasopressors for hepatorenal syndrome (HRS), have been approved [5]. Since treatment on an ICU is highly costly and does often not translate into improved survival rates in these patients, it is of pivotal importance to identify the subgroup of patients for whom further ICU treatment should be omitted and palliative care should be initiated. Initial results of the CANONIC study demonstrated in a subset of patients that the mortality of patients with a CLIF-C ACLFs ≥ 58 points was over 70% [4]. Additionally, a small study conducted by Cardoso et al. could demonstrate that the 90-days mortality rate of patients treated in an ICU with an ACLF and a CLIF-C ACLFs ≥ 65 was 86% [6]. A larger study including 202 patients with ACLF treated in an ICU by Engelmann et al. showed that a cut-off ≥ 70 points may be ideal to identify patients who do not benefit from further ICU treatment [7]. However, before implementation of these cut-offs into routine clinical practice, it is of pivotal importance to validate these findings in different countries and ethnicities. Additionally, data on the predictive ability of the CLIF-C ACLFs regarding long-term survival in patients treated in an ICU are currently scarce. Therefore, it is the aim of the present study to validate the predictive ability of the CLIF-C OFs and CLIF-C ACLFs regarding the usefulness of ongoing ICU treatment and to investigate their predictive ability regarding 28-days (short-), 90-days (medium-), and 365-days (long-term) mortality in patients with liver cirrhosis treated in an ICU in a large German university hospital.

## 2. Material and Methods

### 2.1. Patients

In this retrospective study, a total of 136 consecutive patients with acute decompensation (AD) of liver cirrhosis or ACLF with need for ICU treatment were included between February 2013 and December 2014 at the University Medical Center of the Johannes Gutenberg University, Mainz, Germany. The inclusion period was chosen depending on the necessary number of cases, especially regarding the long-term mortality. Main exclusion criteria were acute liver failure without signs of liver cirrhosis, presence of hepatocellular carcinoma (HCC), or a history of liver transplantation (Figure 1). Data were extracted from the electronic medical records and included demographics, disease characteristics, and laboratory data. The diagnosis of liver cirrhosis was made by ultrasound, CT/MRI scan, histologically or clinically by signs of decompensation, or portal hypertension. The etiology of underlying cirrhosis was divided into alcohol, viral hepatitis, NAFLD, and others. No patients were in intensive care treatment in an external hospital before inclusion in the study.

### 2.2. Definition of ACLF

At ICU admission, the ACLF grade, CLIC-C OFs, and CLIF-C ACLFs were calculated. In addition, the acute decompensation score (ADs) was calculated in patients with decompensated liver cirrhosis not fulfilling the ACLF criteria. The criteria of the EASL-CLIF consortium were used to define acute-on-chronic liver failure [4]. To determine the severity of organ dysfunction in ACLF, the CLIF-organ failure score was calculated, which, in addition to the number of organ failures, measures severity on a numerical scale from a minimum of 30 to a maximum of 90 [4]. The ACLF degree is determined from the number of organ failures and the specific type of organ failure. To calculate the ACLF grade, the specifications of the CLIF-C were used: serum bilirubin ≥ 12 mg/dL; kidney failure: serum creatinine ≥ 2 mg/dL or use of hemodialysis; cerebral failure: grade III-IV hepatic encephalopathy (West Haven criteria); coagulation failure: international normalized ratio (INR) ≥ 2.5 and/or platelets < 20.000/µL; circulatory failure: use of vasopressors to treat severe arterial hypotension. Respiratory failure: PaO2/FiO2 ≤ 200 or SpO2/ FiO2 ≤ 214. Type I ACLF (ACLFI) defines the presence of renal failure alone or of any other type of single organ failure if associated to renal dysfunction (serum creatinine between 1.5 and 1.9 mg/dL) and/or cerebral dysfunction (Grade I or Grade II hepatic encephalopathy). Type II ACLF (ACLF II) or type III ACLF (ACLF III) defines the presence of 2 or 3 to 6 organ failures, respectively [2]. 

### 2.3. Follow-Up

All patients were followed-up during their hospital stay and for 28 days after admission to the ICU via electronic medical records review. In case of a discharge from the hospital, patients were routinely followed-up at least every three to six months (e.g., for routine HCC surveillance) in the outpatient department of the Cirrhosis Center Mainz (CCM). Again, follow-up was conducted retrospectively with help of electronic medical records. The composite endpoint of death or liver transplantation was evaluated as the primary endpoint during follow-up. Since patients with HCC were excluded, all patients who had received a liver transplantation had done so due to final hepatic failure and were consequently treated as complete cases (=death). Survival was defined as transplant-free survival.

### 2.4. Ethics

This study was conducted according to the ethical guidelines of the 1975 Declaration of Helsinki (6th revision, 2008). Anonymous electronic medical records review were obtained for research purposes according to German data protection regulations and law. Accordingly, no informed consent was obtained for observational studies that analyzed anonymized data with no identifiable attributes. 

### 2.5. Statistical Analysis

Quantitative data are expressed as medians with interquartile ranges (IQR). Pairwise comparisons for quantitative variables were performed with an unpaired *t*-test or with the Mann–Whitney U-test. Categorical variables are given as frequencies and percentages, respectively. For the comparison of two or more patient groups, a chi-square test was applied. Regarding the endpoint of death or liver transplantation, survival curves were analyzed using Kaplan–Meier curves and log-rank test. The differences between patients who deceased within different time points (28-days, 90-days, and 365-days) or who survived were assessed by univariate analyses. Variables who reached significance (*p* < 0.05) were subsequently included into multivariable logistic regression models. To investigate the performance of CLIF C OFs, CLIF-C ACLFs, and ADs to identify patients who deceased during the next 28-, 90-, or 365-days, we calculated the area under the receiver operating characteristic curve (AUROC) and its respective 95% confidence interval (95% CI). The ROC analysis is used to measure analysis strategies, in this case the prediction of mortality using the established score. For visual representation, the area under the curve can be calculated as an integral (AUROC). Our complete data analysis is exploratory; hence, no adjustments for multiple testing were performed. For all tests, a 0.05 level was used to define statistically relevant deviations from the respective null hypothesis. However, due to the large number of tests, *p*-values should be interpreted with caution and in connection with effect estimates. Data were analyzed using IBM SPSS Statistic Version 23.0 (IBM Corp., Armonk, NY, USA).

## 3. Results

### 3.1. Patient Characteristics

A total of 189 consecutive patients were screened for this retrospective study. In total, 53 patients were excluded and 136 patients were included into the final analysis (Figure 1). Of those, 19 patients with liver cirrhosis presented with AD and 117 patients presented with ACLF at ICU admission. All patients were followed for at least 28 days. Additionally, 127 patients could be followed up for up to 90 or 365 days, respectively. 

The majority of patients were male (66%) with a median age of 60 (IQR 51; 67). The most common etiology of underlying liver disease was chronic alcohol consumption (65%). The most common grade of ACLF was ACLF III (48%), followed by ACLF II (25%), and ACLF I (13%). At ICU admission, median CLIF-C OFs was 11 (IQR 9; 13) and median CLIF-C ACLFs was 56 (IQR 51; 66) in the entire cohort. In total, 75 patients (55%) were deceased within 28 days after ICU admission. The most common causes leading to decompensation were bleeding, followed by infection, and after-surgical procedures. In total, 55 patients with alcoholic liver cirrhosis showed continued alcohol consumption on admission to hospital or alcohol consumption in the last 6 months before admission to hospital. In total, 13 out of 136 patients with liver cirrhosis were on the waiting list for liver transplantation at the time of inpatient admission. A total of 136 patients were included in the study. Within 28 days, 67 patients died and 8 patients received a liver transplant. In the observation period of 90 days, 12 additional patients died and 3 patients also received a liver transplant. In the observation period up to 365 days, 5 additional patients died and 1 additional patient received a liver transplant. In total, 4 patients could not be followed up more specifically. After study inclusion, 54 patients survived the period of intensive care unit stay, of which 29 died in the maximum observation period after discharge of ICU. All relevant baseline characteristics of the cohort are displayed in Table 1.

### 3.2. Predictors of 28-Days Mortality

In total, 75 patients died or required liver transplantation during follow-up (28-days). In univariable analyses, deceased patients had a higher CLIF-C ACLFs (61 vs. 53, *p* < 0.001) as well as a higher CLIF-C OFs (12 vs. 10, *p* < 0.001) at ICU admission when compared to patients who survived the first 28 days. To identify predictors for higher short-term mortality (28-days) in the entire group as well as in the ACLF subgroup, we conducted different logistic regression analyses including univariable significant variables (Appendix A). In the entire cohort, higher CLIF-C OFs (OR 1.305, 95% CI 1.115–1.527, *p* = 0.001) and CRP (OR 1.007, 95% CI 1.001–1.013, *p* = 0.034) remained significantly associated with higher short-term mortality (Appendix A). In the subgroup of patients with ACLF at ICU admission, CLIF-C ACLFs (OR 1.095, 95% CI 1.041–1.151, *p* < 0.001) was significantly associated with higher short-term mortality, while platelets and CRP did not reach significance (Appendix A). 

To analyze the diagnostic performance of the CLIF-C OFs, CLIF-C ACLFs, and ADs to predict mortality during the 28-days follow-up after ICU admission, ROC curves were conducted. In the entire cohort, the AUROC of the CLIF-C OFs was 0.687 (95% CI 0.599–0.774). In the subgroup of patients with ACLF, the respective AUROCs were 0.652 (95% CI 0.554–0.750) and 0.717 (95% CI 0.626–0.809) for CLIF-C OFs and CLIF-C ACLFs, respectively. The ADs performed well in the subgroup of patients without ACLF at ICU admission with an AUROC of 0.792 (95% CI 0.560–1.000) (Table 2). 

To further analyze the usefulness of the CLIF-C OFs and the CLIF-C ACLFs to predict the futility of further ICU treatment, we calculated the respective sensitivities, specificities, positive predictive values, and negative predictive values regarding 28-days mortality for different cut-offs. These results are displayed in Table 3.

### 3.3. Predictors for Long-Term (90-Days and 365-Days) Mortality

To analyze the usefulness of the CLIF-C O0Fs and the CLIF-C ACLFs to predict 90-days and 365-days mortality, additional analyses were conducted. To identify predictors for higher 90-days and 365-days mortality in the entire group as well as the ACLF subgroup, we conducted different logistic regression analyses including univariable significant variables (Appendix A). In the entire cohort, higher CLIF-C OFs (90-days: OR 1.336, 95% CI 1.106–1.615, *p* = 0.003; 365-days: OR 1.283, 95 CI 1.055–1.561, *p* = 0.012) remained significantly associated with 90-days and 365-days mortality (Appendix A). In the subgroup of patients with ACLF at ICU admission, CLIF-C ACLFs (90-days: OR 1.092, 95% CI 1.025–1.164, *p* = 0.007; 365-days: OR 1.079, 95 CI 1.009–1.154, *p* = 0.027) was significantly associated with 90-days and 365-days mortality (Appendix A). To analyze the diagnostic performance of the CLIF-C OFs, CLIF-C ACLFs, and ADs to predict mortality within 90- or 365-days after ICU admission, ROC curves were conducted. The respective AUROCs are displayed in Table 2.

### 3.4. Survival Analysis in ACLF Grade 3 According to CLIF-C ACLFs

To represent survival, stepwise differentiation was performed based on the ROC analysis and average mortality level using the CLIF-C ACLFs. Logistic regression analysis was performed using conditional forward variable selection to find the optimal model with the respective variables. The logistic regression was performed to find the predictors for transplant-free survival. At a cut-off of 70, a significant difference was shown with respect to short-term 28-days mortality (*p*-Value < 0.001). Over 80% of patients with a CLIF-C ACLFs ≥ 70 at ICU admission died within 28 days (Figure 2). Of the 117 patients who presented with ACLF during ICU treatment, 54 patients were discharged from ICU. The mean time of follow up after discharge from ICU was 593 days (±447 days). After discharge of ICU, 29 patients died or underwent liver transplantation during follow up. Interestingly, neither the maximum ACLF stage nor any treatment was associated with a lower survival (data not shown). In total, six patients underwent liver transplantation after discharge form ICU. Of the patients who survived ACLF, three were transplanted within 28 days, and three patients later than 1 year after ACLF (data not shown). 

## 4. Discussion

In this study, we demonstrated that the predictive ability of the MELD score, the CLIF-C OFs, and the CLIF-C ACLFs was relatively low to predict short- (28-days), medium- (90-days), and long-term (365-days) mortality in patients with ACLF and concomitant need for ICU treatment. However, we found that the CLIF-C ACLFs may have a special merit to judge the futility of further ICU treatment in these patients. In this context, a cut-off of 70 points seems to identify patients with ACLF in whom further ICU treatment should be critically discussed if liver transplantation is not an option. In patients without ACLF, we found that the ADs and MELD score had a slightly better discriminative ability to predict short-term mortality, while the predictive ability regarding long-term mortality decreased remarkably. 

ACLF is a life-threatening complication in patients with decompensated liver cirrhosis, characterized by organ failure and systemic inflammation [8]. In the present study, 90.4% of all consecutive patients with liver cirrhosis admitted to ICU suffered from ACLF as defined by the EASL-CLIF C criteria. Of those, 75 patients deceased within 28 days after ICU admission underlining the poor prognosis of patients with decompensated liver cirrhosis and ACLF with concomitant need for ICU treatment. 

First of all, our results are in line with those of other studies. A meta-analysis by Weil et al. showed a short-term mortality rate of 50% depending on patient selection. Only about one-third of patients were still alive after one year [9,10,11]. In another large study, a comparable reduction in survival was found, with 8–21% patients dying shortly after ICU discharge. In the ICU, mortality rates from 28 days to 1 year were between 47% and 77%, respectively [10]. These data emphasize that causal treatment strategies for this life-threatening condition are urgently needed. In this setting, potential treatment strategies with promising results are, e.g., artificial organ support systems [11] or treatment with granulocyte-colony stimulating factor [12].

The assessment of prognoses in patients with ACLF is complex. Impaired liver function is one of the main determinants of poor prognoses, but patients’ prognoses are also limited by extrahepatic organ failure. Keeping this in mind, the MELD score is used for liver allocation, thus reflecting both liver and renal function [13]. To better stratify patients’ prognosis, the CLIF-C developed and validated the CLIF-C OFs and the CLIF-C ACLFs [2]. Both scores emphasize not only liver function but also extrahepatic organ failure. In this context, it is an interesting finding that the diagnostic accuracy of the CLIF-C OFs and the CLIF-C ACLFs was relatively low in our cohort of patients with concomitant need for ICU treatment. However, this finding is more or less in line with other studies investigating patients with ACLF treated on an ICU (6–7). Though, as an example, in the study conducted by Engelmann et al., the predictive ability of the CLIF-C ACLF score regarding 28-days mortality was slightly better than in our study with an AUC of 0.80, respectively. This may be explained by different time points regarding data collection of the score. Engelman et al. determined the score within the first 48 h after ICU admission, while we determined the score in our patients immediately after ICU admission. However, the results of both studies indicate that the assessment of prognosis in patients with ACLF and concomitant need for ICU treatment remains difficult despite the availability of established scores. This is also reflected by the relatively poor prediction of long-term prognosis by the aforementioned scores in our cohort. We can only hypothesize regarding potential explanations of our findings. ACLF is characterized by high systemic inflammation, which may be at its pinnacle in patients with a need for ICU treatment [14]. It is most likely that a prognosis of these patients relies on additional factors during the first days of ICU treatment that cannot be captured reliably by the established scores. This is additionally supported by the decreasing predictive ability of these scores regarding long-term prognosis. Nevertheless, the CLIF-C ACLFs may have a special merit to judge the futility of further ICU treatment. As also supported by other studies, a cut-off of ≥70 points seems to identify patients with ACLF in whom further ICU treatment should be critically discussed if liver transplantation is not an option [7]. Although liver transplantation is a suitable treatment option for patients with ACLF [15], we were able to show that liver transplantation can only be performed in a small proportion of patients due to various reasons. In our cohort, 13 patients were already on the waiting list for liver transplantation at the time of ICU admission. However, only 12 out of all patients were finally treated by liver transplantation, which was mainly due to the lack of alcohol abstinence > 6 months or due to infections, which often have led itself to the decompensation of liver cirrhosis.

Established prognostic tools such as the MELD score provide the benchmark for all new scores. Our data show that the MELD score has still the best predictive ability of all scores examined in our study. Especially in patients with acute decompensated liver cirrhosis without ALCF, the MELD score showed a very good performance. In contrast, the CLIF-C OFs and the CLIF-C ACLFs performed worse in our study. A retrospective study from Thailand with over 600 patients with acute decompensation of liver cirrhosis was able to provide similar data [16]. Here, the CLIF-OF score also showed only a moderate predictive value in relation to the 30- and 90-day mortality. In contrast to our results, there were differences in the composition of the patients studied. In the Asian collective, considerably more patients with viral liver cirrhosis were treated. In addition, we were able to validate the CLIF-C scores in an intensive care patient population, which is visible in the higher number of advanced ALCF. There is also a lack of data on short-term survival, which is needed to be able to make a suitable therapy decision. However, our data also differ from other comparable studies. Engelmann et al. demonstrated a significantly better prediction of short-term mortality for the CLIF-C OFs and CLIF-C ACLFs in their retrospective analysis [7]. Moreover, the MELD score was inferior to the CLIF-C OFs and the CLIF-C ACLFs. A possible explanation may be that the patients in our study were significantly older and more severely ill. In addition, as already mentioned, the scores were calculated within the first 48 h, and not within the first 24 h as in our study. Possibly, the CLIF-C OFs and the CLIF-C ACLFs loose prognostic accuracy in more ill patients. Moreover, the exact timepoint of data collection seems to be crucial regarding the predictive accuracy. 

The ADs is used for risk stratification in patients with decompensated liver cirrhosis without ACLF. The ADs was originally developed for the general assessment regarding the need for ICU treatment. Based on the prospectively collected data, the authors of the CANONIC study recommend ICU treatment from an ADs of ≥30 [17]. In our study, the ADs showed the best predictive value, with a similar performance as the MELD score. Similar data were collected in a Canadian study [18] and in patients with esophageal variceal bleeding and decompensated liver cirrhosis [19]. These results are also shown by the validation studies of the CLIF consortium. Here, the optimal predictive value was shown particularly in severely ill patients. These results are consistent with our data, with the ADs having the best predictive value in severely ill patient together with the MELD score [4].

One of the major strengths of our study is the inclusion of a large cohort of patients with ACLF at a tertiary care center with a large liver transplantation database. However, there are also limitations that have to be acknowledged. First, we conducted a retrospective study. Second, our study was conducted at a single-center; therefore, the findings may not be generalizable to all patients with liver cirrhosis. Lastly, the group of patients without ACLF and concomitant need for ICU treatment was very small. Therefore, our results are more descriptive and should be interpreted with caution.

In conclusion, we were able to demonstrate that the predictive ability of the MELD score, the CLIF-C OFs, and the CLIF-C ACLFs to predict short- (28-days), medium- (90-days), and long-term (365-days) mortality in patients with ACLF with concomitant need for ICU treatment was relatively low. Thus, we were able to show that the established scores have a low predictive value in intensive care unit patients. However, we found that the CLIF-C ACLFs may have a special merit to judge the futility of further ICU treatment. In this context, a cut-off of ≥70 points seems to identify patients with ACLF in which further ICU treatment should be critically discussed if liver transplantation is not an option.

## 5. Declaration

### Availability of Supporting Data

Data supporting the conclusions of this article is included in the article. Any queries regarding these data may be directed to the corresponding author. 

## Figures and Tables

**Figure 1 medicina-59-00866-f001:**
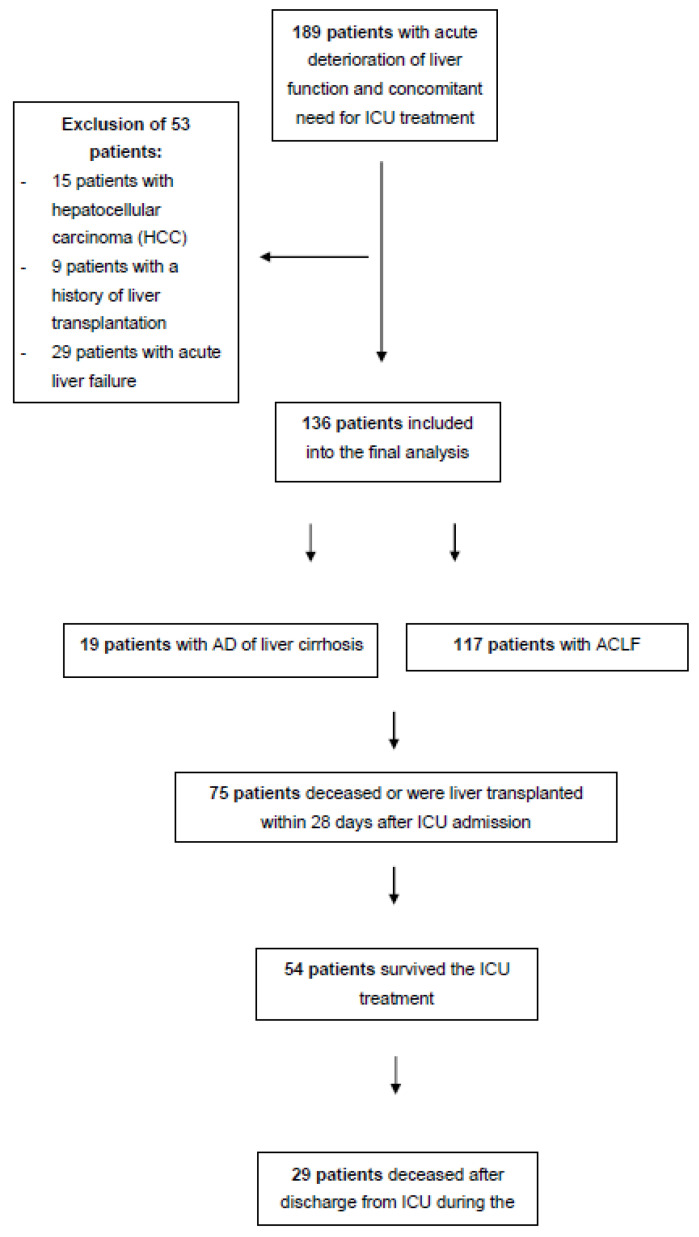
A total of 189 consecutive patients were assessed and after exclusion of 53 patients, 136 patients were considered for the final analysis. A total of 19 patients presented with acute decompensation (AD) of liver cirrhosis and 117 with acute-on-chronic liver failure (ACLF). A total of 75 patients deceased within 28 days after ICU admission. A total of 54 patients survived the ICU treatment and were discharged from ICU. A total of 29 patients deceased after discharge from ICU during the follow up.

**Figure 2 medicina-59-00866-f002:**
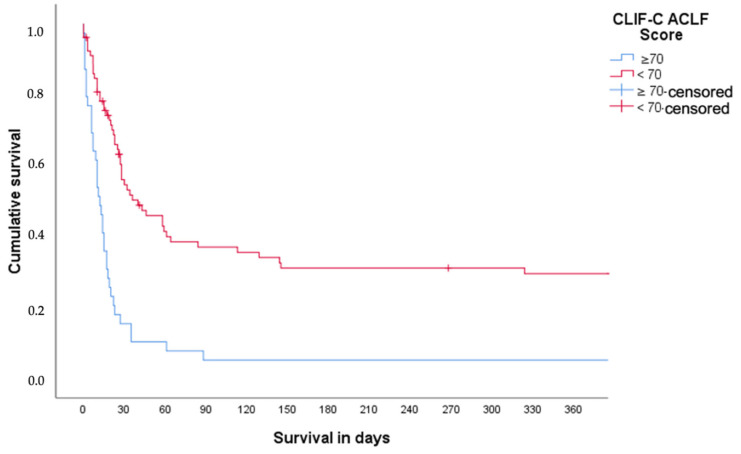
Impact of chronic liver failure consortium (CLIF-C) acute-on-chronic liver failure (ACLF) score with a cut-off 70 regarding the risk for the composite endpoint of death or need for liver transplantation in patients with liver cirrhosis and concomitant need for ICU treatment (*p*-Value < 0.001).

**Table 1 medicina-59-00866-t001:** Demographics and clinical characteristics of the entire cohort and of subgroups stratified by 28-days survival. Data are expressed as medians and IQRs or as frequencies and percentages. NAFLD—nonalcoholic fatty liver disease; ACLF—acute-on-chronic liver failure; MELD—model of endstage liver disease; ICU—intensive care unit; CLIF-C—chronic liver failure consortium; OF—organ failure; WBC—white blood cells; CRP: C—reactive protein.

Variable		All Patients	Deceased within 28 Days	Alive after 28 Days	*p*-Value
N		136	75	61	
Age (years) (IQR)		60 (51; 67)	59 (51; 66)	61 (52; 69)	0.255
Male gender, n (%)		90 (66%)	46 (61%)	44 (72%)	0.186
Etiology	alcohol, n (%) viral hepatitis, n (%) NAFLD, n (%) others, n (%)	89 (65%) 11 (8%) 10 (7%) 17 (13%)	51 (68%) 7 (9%) 6 (8%) 11 (15%)	47 (77%) 4 (7%) 4 (7%) 6 (15%)	0.855
reason for decompensation of liver cirrhosis and ICU admission	Bleeding, n (%) Infection, n (%) after surgical procedure, n (%) Others, n (%)	41 (30%) 27 (20%) 18 (13%) 50 (37%)	20 (27%) 9 (12%) 14 (19%) 32 (42%)	14 (23%) 13 (21%) 6 (10%) 28 (46%)	0.873
ACLF at admission, n (%) - no ACLF - ACLF I - ACLF II - ACLF III		19 (14%) 18 (13%) 34 (25%) 65 (48%)	6 (8%) 10 (13%) 17 (23%) 42 (56%)	13 (21%) 9 (15%) 17 (28%) 23 (38%)	0.072
MELD (IQR)		28 (20; 36)	33 (27; 36)	21 (16; 28)	<0.001
Infection at ICU admission, n (%)		38 (28%)	21 (28%)	17 (28%)	0.975
Renal replacement therapy during ICU treatment, n (%)		78 (57%)	58 (77%)	20 (33%)	<0.001
Mechanical ventilation during ICU treatment, n (%)		75 (55%)	50 (67%)	25 (41%)	<0.003
Vasopressor therapy during ICU treatment, n (%)		117 (86%)	69 (92%)	48 (79%)	<0.026
CLIF-C OF score (IQR) CLIF-C ACLF score (IQR)		11 (9; 13) 56 (51; 66)	12 (10; 14) 61 (53; 71)	10 (8; 12) 53 (49; 58)	<0.001 <0.001
Bilirubin (mg/dL) (IQR)		4 (1.6; 9.9)	6.7 (2.6; 19.6)	2.2 (1.1; 4.8)	<0.001
Sodium (mmol/L)(IQR)		136 (131; 141)	137 (131; 140)	136 (131; 141)	0.613
Platelets (/nL) (IQR)		94 (53; 165)	88 (51; 144)	114 (67; 177)	0.035
INR (IQR)		1.6 (1.3; 2.1)	1.9 (1.5; 2.2)	1.3 (1.2; 1.8)	<0.001
Creatinine (mg/dL) (IQR)		2.1 (1.3; 3.2)	2.4 (1.4; 3.3)	1.9 (1.1; 3.0)	0.167
WBC (/nL) (IQR)		10 (7; 15.4)	11.7 (8.3; 17)	9 (5.8; 15)	0.051
CRP (mg/L) (IQR)		29 (12; 71)	41 (16; 105)	19 (8; 43)	0.005

**Table 2 medicina-59-00866-t002:** Performance of CLIF-C OF score, CLIF-C ACLF score, AD score, and MELD score to predict 28-, 90-, and 365-days mortality in patients with and without acute-on-chronic liver failure (ACLF). Abbreviations: CLIF-C—chronic liver failure consortium; OF—organ failure; ACLF—acute-on-chronic liver failure; AD—acute decompensation; AUROC—area under the receiver operating characteristics curve; 95% CI—95% confidence interval.

	28-Days Mortality AUROC (95% CI)	AUROC (90-Days Mortality)	AUROC (365-Days Mortality)
CLIF_C OF score (total cohort)	0.687 (0.599–0.774)	0.713 (0.613–0.812)	0.689 (0.581–0.796)
CLIF-C OF score (patients with ACLF)	0.652 (0.554–0.750)	0.666 (0.551–0.781)	0.678 (0.552–0.804)
CLIF-C ACLF score (patients with ACLF)	0.717 (0.626–0.809)	0.710 (0.598–0.821)	0.675 (0.550–0.800)
AD score (patients without ACLF)	0.792 (0.560–1.000)	0.713 (0.527–0.899)	0.653 (0.425–0.880)
MELD score (total cohort)	0.795 (0.719–0.870)	0.837 (0.756–0.918)	0.756 (0.655–0.857)
MELD score (patients with ACLF)	0.767 (0.680–0.854)	0.844 (0.758–0.931)	0.784 (0.679–0.890)
MELD score (patients without ACLF)	0.851 (0.657–1.000)	0.722 (0.480–0.965)	0.494 (0.222–0.766)

**Table 3 medicina-59-00866-t003:** Performance of CLIF-C OF score and CLIF-C ACLF score to predict 28-days mortality with different cut-offs. Abbreviations: CLIF-C—chronic liver failure consortium; ACLF—acute-on-chronic liver failure; OF—organ failure; PPV—positive predictive value; NPV—negative predictive value. * measured in 117 patients.

	Sensitivity	Specificity	Positive Predictive Value (PPV)	Negative Predictive Value (NPV)
CLIF-C ACLF score * - ≥70 points - ≥65 points - ≥60 points	25.7 38.6 54.3	97.8 87.2 85.1	94.7 81.8 84.4	6.9 48.8 55.6
CLIF-C OF score - ≥15 points - ≥14 points - ≥13 points	19.7 34.2 39.5	96.7 95.1 86.9	88.2 89.7 78.9	49.2 53.7 53.5

## Data Availability

The authors confirm that the data supporting the findings of this study are available within the article or its Appendix A.

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
