# Peer review of "Validation of the CLIF-C OF Score and CLIF-C ACLF Score to Predict Transplant-Free Survival in Patients with Liver Cirrhosis and Concomitant Need for Intensive Care Unit Treatment"

_medicina, 2023, doi:10.3390/medicina59050866_

Round 1
Reviewer 1 Report
In this manuscript by Michael Nagel and colleagues, the authors reported the ability of CLF-C OF score and CLIF-C ACLF score in patients with liver cirrhosis. While this report contains important information in real world clinical practice, there are several concerns regarding this study that the authors need to clarify.
Major comments
1. As the authors explained in the paper, there are already some studies that validated CLIF-C OF and CLIF-C ACLF scores in patients with ACLF. For instance, this paper from Thailand showed similar results with the present study. The authors should add this in the discussion and emphasize the novelty of this paper.
Validation of prognostic scores predicting mortality in acute liver decompensation or acute-on-chronic liver failure: A Thailand multicenter study | PLOS ONE
2. Currently, there are three diagnostic criteria for ACLF (European Association for the Study of the Liver-Chronic Liver Failure [EASL-CLIF] consortium, North American Consortium for the Study of End-Stage Liver Disease, and Asian Pacific Association for the Study of the Liver). The authors should clarify in the method section that this study was based on the criteria proposed by EASL-CLIF consortium in the method section. In addition, since this paper focused on the CLIF-C OF and CLIF-C ACLF scores, the authors should explain the detail of these scores in the method section with appropriate references.
3. In the method section, the authors explained that the outcome of this study is mortality or liver transplantation. However, the title and manuscript do not reflect this, and this can mislead the readers that the outcome was only mortality. The authors should think of a better word that explains the outcome precisely (transplant-free survival?) and change the explanation throughout the manuscript.
4. Please show the univariate analyses of table 2 and 3 as supplementary files
5. It is difficult to understand the meaning of page 6, lines 226 to 237. In the beginning of this paragraph, there is sentence “To represent survival, stepwise differentiation was performed based on the ROC analysis and average mortality levels using the CLIF-C ACLFs”. I assumed that this analysis also evaluated transplant-free survival not only survival. In addition, what does stepwise differentiation mean? Although I carefully reviewed this manuscript, I could not understand and I think it is the same for the readers. Please reconsider the explanation and clarify the results.
Minor comments
1. In the title, change the word “usefulness” to “validation” because the results show that MELD score is more accurate to predict the outcomes than CLIF-C OF and CLIF-C ACLF scores.
2. Delete the heading in the abstract (Background, Aim, Method, Results, and Conclusion).
3. The authors reported the number of outcomes (death or transplant) in page 5, line 183 and line 209 to 210. Please report the number of death and transplant I detail.
4. The authors reported that “In total, 101 patients died or required liver transplantation during a follow-up of 90 or 365 days”. Does this mean that none of the patients died or received liver transplantation between day 90 and 365 (page 5, line 210 to 211)?
5. Delete the sentences “Interestingly, the time point of 221 the maximum ACLF score also seems to affect long-term mortality. Patients who 222 showed the maximum ACLF score despite at least three days of intensive care treatment 223 were associated with increased mortality (data not shown)” (page 5, lines 221 to 224) and “Interestingly, neither the maximum ACLF stage nor 234 any treatment was associated with a lower survival (data not shown)” (page 5, lines 234 to 235) because these sentences are the interpretation of the authors without showing the results and make the readers confuse.
6. The reference 6 and 7 should be combined in page 7, line 275.
7. It Figure 2, what does 70-zensiert mean? Furthermore, I understand that the curve shows survival or transplant (combined outcome) not survival only. In addition, numbers at risk should be added below the survival curve.
8. Overall, the manuscript contains too many errors and inappropriate English explanations. I recommend the authors to use the English editing service.
Author Response
Point-by-point reply
The revised manuscript has been modified according to the comments raised by both reviewers and all changes in the current version of the manuscript are highlighted in red. We believe that the comments and subsequent changes have significantly improved the quality of our manuscript and after careful revision we hope that the current version can further be considered for publication in Medicina.
Point-by-point reply to all comments raised by the reviewers.
Reviewer #1:
- As the authors explained in the paper, there are already some studies that validated CLIF-C OF and CLIF-C ACLF scores in patients with ACLF. For instance, this paper from Thailand showed similar results with the present study. The authors should add this in the discussion and emphasize the novelty of this paper. Validation of prognostic scores predicting mortality in acute liver decompensation or acute-on-chronic liver failure: A Thailand multicenter study | PLOS ONE
The publication was included in the manuscript and the differences to the collected data were highlighted.
- Currently, there are three diagnostic criteria for ACLF (European Association for the Study of the Liver-Chronic Liver Failure [EASL-CLIF] consortium, North American Consortium for the Study of End-Stage Liver Disease, and Asian Pacific Association for the Study of the Liver). The authors should clarify in the method section that this study was based on the criteria proposed by EASL-CLIF consortium in the method section. In addition, since this paper focused on the CLIF-C OF and CLIF-C ACLF scores, the authors should explain the detail of these scores in the method section with appropriate references.
The use of the EASL - CLIF classification to define the ACLF was named in more detail in the methods section.
- In the method section, the authors explained that the outcome of this study is mortality or liver transplantation. However, the title and manuscript do not reflect this, and this can mislead the readers that the outcome was only mortality. The authors should think of a better word that explains the outcome precisely (transplant-free survival?) and change the explanation throughout the manuscript.
The term survival was replaced by the term transplant-free survival. In addition, the endpoint transplant-free survival was defined and explained in the methods section.
- Please show the univariate analyses of table 2 and 3 as supplementary files
Table 2 and Table 3 have been appended as a supplementary table and the designation in the manuscript has been changed accordingly.
- It is difficult to understand the meaning of page 6, lines 226 to 237. In the beginning of this paragraph, there is sentence “To represent survival, stepwise differentiation was performed based on the ROC analysis and average mortality levels using the CLIF-C ACLFs”. I assumed that this analysis also evaluated transplant-free survival not only survival. In addition, what does stepwise differentiation mean? Although I carefully reviewed this manuscript, I could not understand and I think it is the same for the readers. Please reconsider the explanation and clarify the results.
Logistic regression analysis was performed using conditional forward variable selection to find the optimal model with the respective variables. The logistic regression was performed to find the predictors for transplant – free survival. This information was added to the manuscript.
- In the title, change the word “usefulness” to “validation” because the results show that MELD score is more accurate to predict the outcomes than CLIF-C OF and CLIF-C ACLF scores.
The title was changed in the manuscript.
- Delete the heading in the abstract (Background, Aim, Method, Results, and Conclusion).
The heading in the abstract was deleted.
- The authors reported the number of outcomes (death or transplant) in page 5, line 183 and line 209 to 210. Please report the number of death and transplant I detail.
The number of patients who died and the number of patients with a liver transplant were added to the manuscript and presented in more detail.
- The authors reported that “In total, 101 patients died or required liver transplantation during a follow-up of 90 or 365 days”. Does this mean that none of the patients died or received liver transplantation between day 90 and 365 (page 5, line 210 to 211)
The number of deceased patients and patients with liver transplantation were added and divided between the respective observation periods.
- Delete the sentences “Interestingly, the time point of 221 the maximum ACLF score also seems to affect long-term mortality. Patients who 222 showed the maximum ACLF score despite at least three days of intensive care treatment 223 were associated with increased mortality (data not shown)” (page 5, lines 221 to 224) and “Interestingly, neither the maximum ACLF stage nor 234 any treatment was associated with a lower survival (data not shown)” (page 5, lines 234 to 235) because these sentences are the interpretation of the authors without showing the results and make the readers confuse.
The named sentences were deleted in the results.
- The reference 6 and 7 should be combined in page 7, line 275.
The references were combined.
- It Figure 2, what does 70-zensiert mean? Furthermore, I understand that the curve shows survival or transplant (combined outcome) not survival only. In addition, numbers at risk should be added below the survival curve.
If an outcome is not recorded in its full form and the exact time of the event is not known, this data is indicated as censored. The expression has been translated into English.
- Overall, the manuscript contains too many errors and inappropriate English explanations. I recommend the authors to use the English editing service.
The text has been carefully corrected and revised.

Reviewer 2 Report
This manuscript lacks any scientific contribution. The aim of the authors to investigate already developed and established, well known scores (CLIF-C OFs, CLIF-C ACLFs or any other score that has been present for almost a decade) is futile. Scores, in general, can be the focus of attention and investigation only if they are ORIGINAL (like in ref. 7, 2018; ref. 15, 2015, ref. 18, 2015) or very recently developed or subjected to meta-analysis. Otherwise, they are just tools to help clinicians. Even if there was any need for validation, the sample size in this manuscript is far too small (19 patients with AD and 117 of them with ACLF at ICU admission) for any conclusion whatsoever (especially in terms of subgroup analysis). Sensitivity and specificity of the variables (AUROC) was reported, yet it was done without Youden index which is essential for objective determination. There are ONLY two new RELEVANT references. In ref. 16, there were almost 400 patients, in ref. 17 there were more than 600 patients. All other references are too old; only four more are new but not related to scoring systems. Also, all references are incompletely cited. The manuscript should be rejected.
Author Response
Thank you for the helpful comments and remarks. Certainly, the scores studied have existed for some time, but in clinical care it is difficult to assess the short- and especially the long-term prognosis of patients with ACLF. To improve this situation, we conducted this retrospective study. We deliberately used older scores that are already established in clinical care and are therefore easy to calculate. Besides the scientific importance of this retrospective study, the main aim is to solve clinically relevant problems for the treating physicians. The statistical evaluation was prepared in cooperation with our statistical department. We would be happy to adapt the references.

Reviewer 3 Report
In the manuscript titled “Usefulness of the CLIF-C OF score and CLIF-C ACLF score to predict mortality in patients with liver cirrhosis and concomitant need for intensive care unit treatment” Michael Nagel et al.present a retrospective study with an aim to validate the predictive ability of CLIF-C OFs and CLIF-C ACLFs on the rationale of ICU treatment and short- and long-term mortality. The topic is relevant and interesting. However, there are several points of criticism:
1. The patient cohort is 10 years old and is from 2013-2014. The validation of the scores was performed in an old cohort. Why was this specific time frame chosen? It would be a nice study if all consecutive patients from 2013 would be included.
2. The aim was “to validate”. I’m missing clear statement in the conclusions if authors were able to validate the selected scores.
3. In the abstract and text multiple times term “AUROC” is repeated. It should be clearly explained for what it’s used and what kind of statistical purpose it has.
4. Authors are stating that scores were calculated during the first 24 hours of ICU admission. Were there any patients which were transferred from ICU of other hospitals and the ACLF was suspected/diagnosed already before?
5. 75 patients deceased or were liver transplanted within 28 days after ICU admission and 54 patients survived the ICU treatment. What happened to other 6 patients?
6. Tables visually look confusing. Must be remade.
7. Figure 2 has some German words.
Author Response
Reviewer #3:
In the manuscript titled “Usefulness of the CLIF-C OF score and CLIF-C ACLF score to predict mortality in patients with liver cirrhosis and concomitant need for intensive care unit treatment” Michael Nagel et al. present a retrospective study with an aim to validate the predictive ability of CLIF-C OFs and CLIF-C ACLFs on the rationale of ICU treatment and short- and long-term mortality. The topic is relevant and interesting. However, there are several points of criticism:
- The patient cohort is 10 years old and is from 2013-2014. The validation of the scores was performed in an old cohort. Why was this specific time frame chosen? It would be a nice study if all consecutive patients from 2013 would be included.
An older retrospective cohort was chosen to allow for a sufficiently long follow-up period. The duration of patient inclusion was used to recruit a sufficiently large number of cases. This information was added to the manuscript. Certainly, current data would be of interest and should be investigated in a prospective study.
- The aim was “to validate”. I’m missing clear statement in the conclusions if authors were able to validate the selected scores.
The results on the predictive value of the scores examined were emphasized in the results section.
- In the abstract and text multiple times term “AUROC” is repeated. It should be clearly explained for what it’s used and what kind of statistical purpose it has.
The ROC analysis is used to measure analysis strategies, in this case the prediction of mortality using the established score. For visual representation, the area under the curve can be calculated as an integral (AUROC). This explanation was added to the manuscript.
- Authors are stating that scores were calculated during the first 24 hours of ICU admission. Were there any patients which were transferred from ICU of other hospitals and the ACLF was suspected/diagnosed already before?
No patient was in external intensive care before inclusion in the study. This information was added to the manuscript.
- 75 patients deceased or were liver transplanted within 28 days after ICU admission and 54 patients survived the ICU treatment. What happened to other 6 patients?
75 patients died within 28 days in intensive care, resulting in 61 patients still in intensive care after 28 days. Within the following period, another 6 patients died in intensive medical treatment, so that 54 patients could be discharged from intensive medical treatment. The data were supplemented in the manuscript.
- Tables visually look confusing. Must be remade.
2 tables were declared as supplementary data to generate more overview. The remaining tables were revised.
- Figure 2 has some German words.
Figure 2 has been revised.

Round 2
Reviewer 1 Report
The authors have satisfyingly answered all my comments.
Reviewer 2 Report
I fully DISAGREE with the explanation provided by the authors. The manuscript has not been improved at all and therefore it should be rejected.
Reviewer 3 Report
Thank you for the corrections. I have no other comments.